# COVID-19 related morbidity and mortality in people experiencing homelessness in the Netherlands

**Eline Mennis**[1☯], **Michelle Hobus**[1☯], **Maria van den Muijsenbergh**[1,2], **Tessa van Loenen**[1,2] *

**1** Department of Primary and Community Care, Radboud University Medical Centre, Nijmegen, The Netherlands, **2** Pharos, Centre of Expertise on Health Disparities, Utrecht, The Netherlands

☯ These authors contributed equally to this work.
* Tessa.vanloenen@radboudumc.nl

**Data Availability Statement:** Data cannot be shared publicly because of ethical restrictions. Data that can be shared are available from Dans Easy https://doi.org/10.17026/dans-2c7-wksz. All other

## Abstract

### Introduction

People who are homeless might be more at risk for getting infected by the SARS-COV-2 virus or for experiencing severe course of the infection due to their often more fragile health, unmet health needs, and poorer living conditions. This study aims to gain insight into the morbidity and mortality of the SARS-COV-2 virus among the homeless population in the Netherlands.

### Methods

In this observational retrospective study, anonymized data about patients experiencing homelessness who contacted a street doctor were gathered in nine cities in the Netherlands from March 2020 until March 2021. Data included patient characteristics, COVID-19 -related symptoms, diagnosis, and disease course of a SARS-COV-2 infection.

### Results

Of the total 1419 patients in whom 1544 COVID-19 related consults were registered, 16% tested positive for a SARS-COV-2 infection, and an additional 12% were clinically suspected of having a SARS-COV-2 infection but were seen before there were any COVID-19 tests available in General Practice or for some other reason not tested. Significantly more (p = <0.001) patients born outside the European Union tested positive for a SARS-COV-2 infection (36%) compared to the remainder of the population (20%). The most discerning symptom for being tested positive was loss of taste and smell (29% vs 6% in the negative tested group and 2% in the suspected group), fever (24% vs 13% in the negative tested group but 18% in the suspected group), and headache (26% vs 17% resp 14%), and fatigue (29% vs 20 resp 17%). Cough, dyspnea and common cold were more often seen in the clinically suspected but not tested group). Of the group that tested positive for a SARS-COV-2 infection, 10% were hospitalized. Two patients were admitted to intensive care and one patient died. Of patients who were clinically suspected of a SARS-COV-2 infection, 5% were hospitalized.

data are available on request at the Data Stewards of Radboudumc-ELG (contact via kwaliteitsteam. elg@radboudumc.nl) for researchers who meet the criteria for access to confidential data.

**Funding:** Funding for this study was provided by the Netherlands Organisation for Health Research and Development (ZonMw) (https://www.zonmw. nl/en/), project number 10430022010005. ZonMw had no role in study design, data collection and analysis, decision to publish or preparation of the manuscript.

**Competing interests:** the authors have declared that no competing interests exist.

## Conclusion

COVID-19 was not widespread among people experiencing homelessness in the Netherlands, but the number of hospitalizations in this study was relatively high. Monitoring this population during a pandemic is necessary to take prompt action when needed.

## Introduction

The COVID-19 epidemic, caused by a novel coronavirus named SARS-CoV-2, has become a pandemic since it was first detected in December 2019. On February 27th 2020, the first patient with a SARS-COV-2 infection was confirmed in the Netherlands [1]. Common symptoms of a SARS-COV-2 infection are symptoms of a cold, sore throat, coughing, dyspnea, fever, or a loss of smell or taste. Risk factors for a severe course of the infection appeared to be old age, underlying chronic morbidities (such as hypertension, cardiovascular diseases, diabetes, respiratory diseases, and obesity), or a compromised immune system (e.g. malignancy or HIV) [2, 3]), especially those who are not vaccinated [4]. Ethnic minorities also seem more at risk for a severe course of SARS-COV-2 infection [5, 6], also in the Netherlands [7, 8].

Research on other epidemics shows that infections and the Public health and social measures (PHSM) to contain the spread of the virus have more impact on vulnerable groups like people experiencing homelessness than on the average population [9–11]. Many people who experience homelessness suffer from physical and/or mental chronic diseases [12, 13]. Besides, their living circumstances make them more prone to get infected by SARS-COV-2 or experiencing its societal consequences. An example is the higher risk of transmission of COVID-19 in a homeless shelter; sleeping sites are often very crowded, people move between different cities and shelter locations, implementing protective measures (for example social distancing and use of mouth masks) in shelters is difficult or transmission may be asymptomatic [14]. Moreover, high rates of mental and physical comorbidity [13], drug abuse [15], and unmet care needs [16, 17], could lead to several difficulties in screening, quarantining, and treating people experiencing homelessness with a SARS-COV-2 infection [18]. Also, the reduction in face-to-face health service contact and the cancellation of the outreaching provision of care might lead to a lower propensity to seek care when needed [19]. All these factors made this already vulnerable population in society even more vulnerable during the COVID-19 pandemic. While the pandemic continued, healthcare professionals in different countries all over the world have been expressing their concerns about the efforts governments take to control the COVID-19 outbreak, but leaving out one of the most vulnerable groups in society: the homeless population [12, 18, 20–22].

The number of people experiencing homelessness in the Netherlands doubled in the last decade from 17.800 in 2009 to at least 39.300 in 2018 [23]. It is expected that the long-term social, political, and economic consequences of the COVID-19 pandemic will further increase this number [24]. People experiencing homelessness in the Netherlands are often not registered at a General Practitioner (GP) practice [25]. Yet, outreaching primary healthcare for patients who are homeless is being provided by so called "street doctor practices", which consists of GPs, Public Health doctors, and / or nurses. They see patients experiencing homelessness with health problems at the request of the patients themselves or the staff of homeless shelters [16]. They are often the first entry point for COVID-19 -related care for the homeless population.

To contain the virus, the Ministry of Health, Welfare and Sport (VWS) in the Netherlands issued in March 2020 the first guideline for implementing public health and social measures

(PHSM) in shelters and care facilities for people experiencing homelessness as well as a minimum of conditions in which social and medical care should be organized during the COVID-19 pandemic [26, 27]. This guideline has been frequently updated but kept all the basic measures (e.g. hygiene measures and social distancing). During the several lockdowns, which started in March 2020, October 2020, and December 2021, the Ministry of VWS called on municipalities to open night shelters for all people experiencing homelessness, including those who are not legally entitled to access the shelter, like undocumented migrants [27].

In the first weeks of the pandemic PCR tests to test for the COVID-19 virus were not fully available. In the guideline published on the 18th of May 2020, it was not recommended to test persons with COVID-19-related complaints outside the hospital because this would have no added value for the policy to be pursued [28]. From June 1st 2020 onwards all people in the Netherlands could get tested when they presented symptoms [29, 30].

Since the start of the COVID-19 pandemic, the National Institute of Public health and the Environment (RIVM) daily reported the number of infections, hospital admissions, intensive care admissions, and deaths in the Netherlands. However, people experiencing homelessness are not represented in databases on which the numbers of the RIVM were based. As mentioned, they often are not enlisted with or do not visit a regular GP [25]. Consequently, COVID-19 -related morbidity and primary care of people experiencing homelessness were not monitored with existing GP registration networks, resulting in lacking data on people experiencing homelessness on SARS-COV-2 infections. Therefore, this study aims to gain insight into the morbidity and mortality of COVID-19 among the homeless population. These insights contribute to the body of knowledge on the impact of the COVID-19 pandemic on the health of people experiencing homelessness which can be used to optimize pandemic-proof healthcare for this vulnerable group in our society.

## Methods

### Study design

This study performed an restrospective analysis of medical records of streetdoctors of patients experiencing homelessness visiting a street doctor practice in the five largest and four smaller cities in different parts of the Netherlands: Amsterdam, Rotterdam, Den Haag, Utrecht, Eindhoven, Heerlen, Nijmegen, Tilburg and Almere. The number of people experiencing homelessness living in the cities differs, Amsterdam and Rotterdam having the largest homeless populations in the Netherlands.

In each of the nine cities, a street doctor practice participated. Street doctor practices were recruited in collaboration with the Dutch Street doctor Group (NSG). People experiencing homelessness are defined broadly as people who lack a steady home, and live in emergency shelters, outdoors, or in buildings not meant for shelter. This study includes all persons who use the service of a street doctor practice, whatever their sleeping place is.

The study population included people experiencing homelessness with the following inclusion criteria: 1) age above 16 years, 2) contact with one of the street doctor practices in the nine cities with SARS-COV-2 -related issues, complaints, or questions.

### Data collection

Data was collected from March 1st 2020 until March 1st 2021. Street doctors practices use different systems to record their medical data of patients and some do not have a data system for people who are homeless. Extracting the required data and comparing the data was not possible with the existing systems and processes. For this reason, we created a standardised registration form where street doctors could enter the relevant data from the consultation they had

with each patient. This form was created and finalized in May 2020 using an online survey programme Castor EDC. Items were based on literature available in May 2020 about COVID-19 and the expertise of experienced street doctors. For each patient the street doctor saw, a form was filled in with the required data. The data from March 2020 until May 2020 was were added in retrospect. As of May 2020, the street doctors received a reminder twice a month to enter the data of the patients they had seen in the previous two weeks. The registration form requested data on patient characteristics, COVID-19 -related symptoms, diagnosis, and disease course. Once a month the street doctors were contacted and asked to fill in a questionnaire for each of the patients experiencing homelessness they had seen in their street doctors practice with COVID-19 -related issues, complaints or questions in the past month. As there is no national patient registry and the street doctor practices each have their registration system, the street doctors themselves retrieved the data from their own patient records and transferred them to the online registration form.

## Categorizing by diagnosis

Patients seen by the street doctors were categorized into four groups: PCR-positive for SARS-COV-2, PCR-negative for SARS-COV-2, clinically suspected for SARS-COV-2 but not tested, and clinically not suspected for a SARS-COV-2 infection. Patients were categorized as PCR-positive or PCR-negative when this was confirmed by a positive or negative PCR test. This includes patients with or without complaints like patients who had been in contact with a patient who tested positive for a SARS-COV-2 infection. Patients were categorized as clinically suspected if the street doctor made a diagnosis of a SARS-COV-2 infection but it was not possible to confirm this with a molecular or antigenic test. Especially, at the beginning of the pandemic due to a shortage of PCR tests and policy guidelines, not all people experiencing homelessness who contacted the street doctors could be tested. In addition, some refused to be tested. Street doctors therefore had to base their diagnosis on multiple facts and observations from history and taking physical examination. Those cases, where all symptoms indicated to COVID-19 and doctors could or did not perform a PCR-test, were classified the case as "clinically suspect." And on the other hand, when fever or other complaints were more likely explained by other conditions, they will have classified this as "not-suspected" of SARS-COV-2 infection.

## Data analysis

Statistical analysis was performed using IBM SPSS Statistics for Windows, version 25. Descriptive statistics were used to describe the baseline characteristics of the study population, COVID-19 -related symptoms, diagnosis, and disease course of a SARS-COV-2 infection. Pearson's chi-square tests were used to determine statistical significance.

## Ethical considerations

The study had been approved by the Medical Ethical Committee of the Radboud university medical center (CMO Radboudumc) (CMO) Arnhem-Nijmegen (nr 2020 6428). All data were anonymized before being entered into the on-line questionnaire and therefor individual patient consent was deemed not required.

## Results

### Number of infections

There were a total of 1544 consultations for COVID-19 related complaints or issues distributed among 1419 patients experiencing homelessness. 1305 of the patients came for one

**Table 1. Number of episodes per city between March 2020 and March 2021.**

| City | N |
|------|---|
| Almere | 2 |
| Amsterdam | 741 |
| Den Haag | 104 |
| Eindhoven | 102 |
| Heerlen | 75 |
| Nijmegen | 15 |
| Rotterdam | 344 |
| Tilburg | 98 |
| Utrecht | 63 |
| Total | 1544 |

consultation and 114 patients came more than once (104 patients came 2 times, 9 patients came 3 times and 1 patient came 4 times). The majority of episodes collected, originate from Amsterdam (48%) and Rotterdam (22%), as shown in Table 1.

In 16% of these 1544 episodes, a positive test result for SARS-COV-2 was registered. More than half (56%) of the patients tested negative for SARS-COV-2 despite their complaints, and 12% were clinically suspected of SARS-COV-2 but not tested. 16% of the patients were clinically not suspected of SARS-COV-2 after being assessed by street doctors.

Fig 1 presents the test results per month. Several peaks in infections were seen in different periods, namely in October 2020 (18%), November 2020 (16%), and February 2021 (20%). In May—July 2020 almost no patients tested positive for a SARS-COV-2 infection.

## Characteristics of the study population

Table 2 provides the characteristics of the study population. The median age was 44 years (interquartile range 22), ranging from 17 to 84 years old. They were mostly men (70%). 29% of

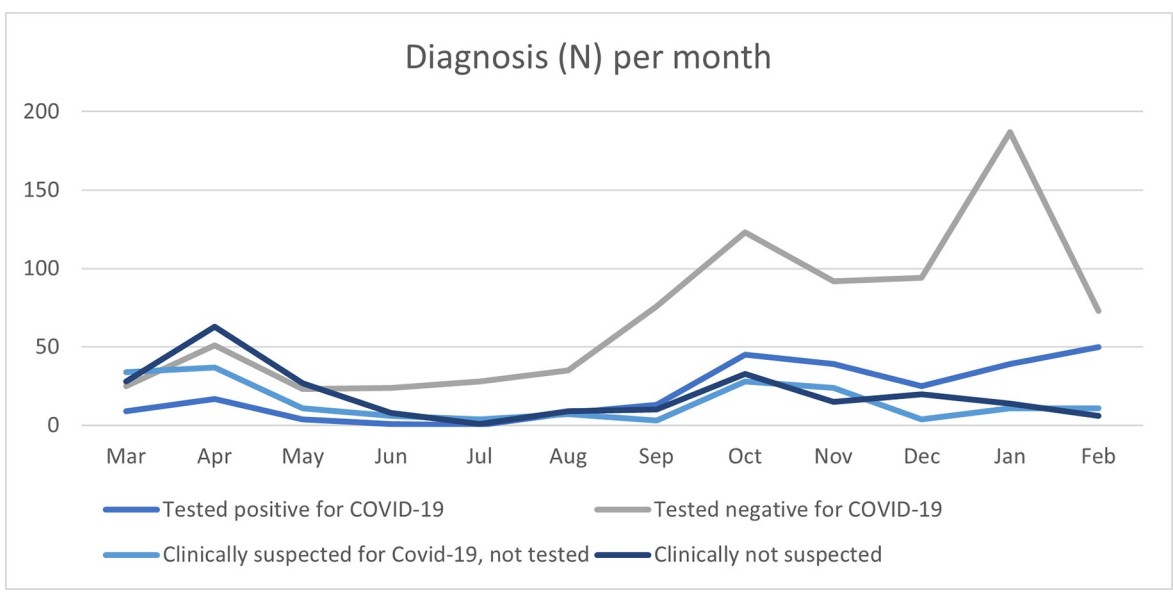

**Fig 1. Diagnosis (N) per month between March 2020 and March 2021.**

**Table 2. Baseline characteristics of study population and diagnosis groups.**

| Characteristics N (%) | | Total study population (N = 1419) * | Tested positive (N = 251) | Tested negative (N = 847) | Clinical suspected (N = 183) | Clinical not suspected (N = 245) |
|---|---|---|---|---|---|---|
| Age Median in years (interquartile range) | | 44 (22) | 43 (22) | 46 (21) | 49 (22) | 42 (22) |
| **Biological sex** | Female | 306 (22) | 53 (21) | 194 (23) | 37 (20) | 53 (22) |
| | Male | 987 (70) | 182 (73) | 601 (71) | 92 (50) | 187 (76) |
| | Unknown | 126 (9) | 16 (6) | 52 (6) | 54 (30) | 5 (2) |
| **Country of Birth** | The Netherlands | 418 (29) | 41 (16) | 321 (38) | 46 (25) | 63 (26) |
| | European Union | 113 (8) | 11 (4) | 83 (10) | 16 (9) | 12 (5) |
| | Outside of the European Union | 411 (29) | 91 (36) | 205 (24) | 43 (23) | 94 (38) |
| | Unknown | 477 (34) | 108 (43) | 238 (28) | 78 (43) | 76 (31) |
| **Health care insurance** | Yes | 751 (53) | 152 (61) | 521 (62) | 63 (34) | 104 (43) |
| | No | 346 (25) | 64 (26) | 180 (21) | 38 (21) | 82 (34) |
| | Unknown | 313 (22) | 34 (14) | 140 (17) | 82 (45) | 57 (23) |
| **Registered by GP** | Yes | 579 (41) | 128 (52) | 378 (45) | 55 (30) | 84 (34) |
| | No | 191 (14) | 43 (17) | 107 (13) | 13 (7) | 40 (16) |
| | Unknown | 635 (45) | 77 (31) | 349 (42) | 115 (63) | 121 (49) |
| **COPD** | Yes | 114 (8) | 21 (8) | 98 (12) | 7 (4) | 11 (4) |
| | No | 637 (45) | 158 (63) | 355 (42) | 71 (39) | 121 (49) |
| | Unknown | 667 (47) | 72 (29) | 394 (47) | 105 (57) | 113 (46) |
| **Diabetes mellitus (type I or/and II)** | Yes | 71 (5) | 19 (8) | 45 (5) | 5 (3) | 9 (4) |
| | No | 685 (48) | 162 (65) | 405 (48) | 72 (39) | 126 (51) |
| | Unknown | 662 (47) | 70 (28) | 397 (47) | 106 (58) | 110 (45) |
| **Hypertension** | Yes | 95 (7) | 25 (10) | 59 (7) | 11 (6) | 14 (6) |
| | No | 641 (45) | 150 (60) | 382 (45) | 65 (36) | 116 (47) |
| | Unknown | 682 (48) | 76 (30) | 406 (48) | 107 (58) | 115 (47) |
| **Cardiovascular disorders (other than hypertension)** | Yes | 75 (5) | 17 (7) | 49 (6) | 7 (4) | 13 (5) |
| | No | 652 (46) | 158 (63) | 386 (46) | 66 (36) | 118 (48) |
| | Unknown | 691 (49) | 76 (30) | 412 (49) | 110 (60) | 114 (47) |

* Total of 1419 unique participants, 114 people contacted the street doctors for two or more episodes

the patients were born in the Netherlands and 29% were born outside the European Union. About one-third (30%) was previously known by street doctors.

As for the risk factors for a severe course of a SARS-COV-2 infection in the positive tested group, 8% had COPD, 8% had diabetes mellitus and 10% had hypertension, which was slightly more compared to the total study population. Significantly more (p = <0.001) patients born outside the European Union tested positive for SARS-COV-2 (36%) compared to the remainder of the population (20%). Registration of patient data in street doctor databases was often incomplete, resulting in a relatively high percentage unknowns, remarkably often in the group "clinical suspected (and not tested)".

## Complaints and symptoms

In the group that tested positive for SARS-COV-2 the most common symptoms were a cold (40%) and a cough (45%). Nevertheless, these symptoms were also the most common in the group that tested negative for SARS-COV-2. Loss of taste and smell (29%), fever (24%) headache (26%), and fatigue (29%) were more present in the group that tested positive for SARS-COV-2 than in the group that tested negative, see Table 3.

**Table 3. Symptoms per diagnosis group.**

| Symptoms N (%) | | Tested positive (N = 251) | Tested negative (N = 847) | Clinical suspected (N = 183) | Clinical not suspected (N = 245) |
|---|---|---|---|---|---|
| **Catch a cold, sniffle, running nose** | Yes | 82 (40) | 385 (52) | 83 (66) | 8 (3) |
| | No | 108 (53) | 326 (44) | 38 (30) | 216 (94) |
| | Unknown | 14 (7) | 27 (4) | 4 (3) | 5 (2) |
| **Sneezing** | Yes | 21 (10) | 78 (11) | 14 (11) | 1 (0) |
| | No | 168 (82) | 625 (85) | 104 (84) | 225 (98) |
| | Unknown | 15 (7) | 36 (5) | 6 (5) | 3 (1) |
| **Loss of smell and/or taste** | Yes | 54 (29) | 44 (6) | 1 (2) | 0 (0) |
| | No | 112 (60) | 574 (82) | 29 (56) | 104 (58) |
| | Unknown | 21 (11) | 81 (11) | 22 (42) | 75 (42) |
| **Sore throat** | Yes | 60 (29) | 203 (28) | 43 (35) | 7 (3) |
| | No | 130 (64) | 500 (68) | 78 (63) | 218 (95) |
| | Unknown | 14 (7) | 34 (4) | 3 (2) | 4 (2) |
| **Cough** | Yes | 84 (45) | 339 (48) | 28 (54) | 7 (4) |
| | No | 86 (46) | 318 (45) | 19 (37) | 163 (91) |
| | Unknown | 17 (9) | 42 (6) | 5 (10) | 9 (5) |
| **Shortness of breath** | Yes | 43 (21) | 153 (21) | 41 (33) | 2 (1) |
| | No | 150 (74) | 549 (74) | 80 (65) | 223 (97) |
| | Unknown | 11 (5) | 36 (5) | 3 (2) | 4 (2) |
| **Fever (body temperature > 38 Celsius)** | Yes | 48 (24) | 98 (13) | 35 (28) | 10 (4) |
| | No | 142 (70) | 604 (82) | 83 (67) | 213 (93) |
| | Unknown | 14 (7) | 37 (5) | 6 (5) | 6 (3) |
| **Headache** | Yes | 54 (26) | 126 (17) | 17 (14) | 4 (2) |
| | No | 136 (67) | 575 (78) | 102 (82) | 221 (97) |
| | Unknown | 14 (7) | 37 (5) | 5 (4) | 4 (2) |
| **Fatigue** | Yes | 59 (29) | 145 (20) | 21 (17) | 1 (0) |
| | No | 133 (65) | 532 (72) | 92 (74) | 167 (73) |
| | Unknown | 12 (6) | 62 (8) | 11 (9) | 61 (27) |
| **Nausea** | Yes | 18 (9) | 52 (7) | 8 (6) | 0 (0) |
| | No | 173 (85) | 640 (87) | 110 (89) | 225 (98) |
| | Unknown | 13 (6) | 47 (6) | 6 (5) | 4 (2) |
| **Stomach ache** | Yes | 11 (6) | 24 (3) | 2 (4) | 0 (0) |
| | No | 163 (87) | 629 (90) | 44 (85) | 175 (98) |
| | Unknown | 13 (7) | 46 (6) | 6 (12) | 4 (2) |
| **Diarrhea** | Yes | 9 (5) | 24 (3) | 2 (4) | 0 (0) |
| | No | 164 (88) | 629 (90) | 44 (85) | 175 (98) |
| | Unknown | 14 (7) | 46 (6) | 6 (12) | 4 (2) |

The most common symptoms of patients in the group clinical suspected of COVID-19 were a cold (66%), a cough (54%), a sore throat (35%), and shortness of breath (33%). It was remarkable that in the group of patients that were clinically suspected of COVID-19 loss of taste and smell only occurred in 2%, whereas it was more common in the group that tested positive for SARS-COV-2 (29%). Fever was reported more in the group 'clinically suspected" (28%) than in the group was that tested negative for SARS-COV-2 (13%).

## Disease course

The percentage of people experiencing homelessness admitted to the hospital was higher among the group that tested positive for SARS-COV-2 (10%) and clinically suspected of

**Table 4. Disease course per group between March 2020 and March 2021.**

| N (%) | Tested positive (N = 251) | Tested negative (N = 847) | Clinical suspected (N = 183) | Clinical not suspected (N = 245) |
|---|---|---|---|---|
| **Admitted to a hospital ward** | 24 (10) | 16 (2) | 9 (5) | 0 (0) |
| **Admitted to an intensive care unit** | 2 (1) | 2 (0) | 0 (0) | 0 (0) |
| **Passed away** | 1 (0) | 0 (0) | 0 (0) | 0 (0) |

SARS-COV-2 (5%) than in the group that tested negative for SARS-COV-2 (2%). None of the patients who were clinically suspected of a SARS-COV-2 infection were admitted to the ICU and neither did any of those patients pass away, see Table 4.

The positive tested patients who were admitted to the hospital (N = 24) were overall men (75%) with a median age of 52 (interquartile range 21) of whom 56% were born outside the European Union. In this group more symptoms were registered: cough (75%), fatigue (59%), loss of smell and taste (50%), shortness of breath (50%) and fever (45%). One patient with a migration background who tested positive for a SARS-COV-2 infection was admitted to the intensive care unit and passed away.

## Discussion

### Main findings

To our knowledge, this is the first study on COVID-19 -related morbidity among people experiencing homelessness in the Netherlands. 1419 patients in nine cities contacted street doctor practices with COVID-19 -related issues between March, 1st 2020 and March, 1st 2021. Of them, 16% tested positive for SARS-COV-2, and in addition 12% were clinically suspected of SARS-COV-2. These numbers were lower than anticipated given the vulnerability and increased exposure of people experiencing homelessness. Based on these numbers there seems to be no indication that COVID-19 among the group of people experiencing homelessness led to more morbidity than among peers, however we realize that our methods do not allow a conclusion on this comparison. Worldwide data published on prevalence rates of SARS-COV-2 infections among homeless population vary among different shelters and different countries, with an overall mean prevalence of 32% [31]. The number of SARS-COV-2 infections in our study population seems to be lower. This could be explained by the early implementation of the preventive measures, including hygiene, isolation of suspected or positive tested patients, reducing the number of beds at the dorms, and setting up emergency locations at the beginning of the COVID-19 pandemic [26, 27]. The peaks of infections we registered coincide with the national course of infection rates [32].

The symptoms registered in the positive tested group are also comparable to those of the COVID-19 patients in the general population in the Netherlands [33]. The significantly higher number of migrants from outside the European Union who tested positive for SARS-COV-2 is also in line with other studies [5–7]. The reason for this difference remains not fully understood. For migrants groups in the general population these disparities have been related to socioeconomic deprivation, differences in household composition, occupations with higher risk of infection and increased exposure to crowded conditions, in combination with barriers in accessing healthcare [8]. However, these factors are not likely to be much different between migrant and other people experiencing homelessness. Two possible explanations could be the lack of appropriate communication for migrants [34] as the effect of racial discrimination that is often experienced among migrants [35, 36].

The percentage of patients admitted to the hospital for a SARS-COV-2 infection in the study population (10%) is higher than that in the general population in the Netherlands,

according to the National Institute of Public Health and the Environment (RIVM) in the same period March 2020 to February 2021, which was approximately 4 percent [37, 38]. This is an indication of the high prevalence of risk factors for severe course of aSARS-COV-2 infection at a younger age in people experiencing homelessness compared to other people: Although the age of the group COVID-19 patients admitted to the hospital was higher than the mean age of the total study population (52 versus 44 years), confirming that higher age is a risk factor for severe SARS-COV-2 infection [36], this age is substantially lower than the mean age of other Dutch patients admitted to the hospital with a SARS-COV-2 infection (67 years) [37].

## Strengths and limitations

This study is the first and only epidemiological study on COVID-19 -related morbidity among people experiencing homelessness e in the Netherlands in the first year of the pandemic. They are a challenging group to capture in epidemiological data because of the mobile and variable nature of this group. They are hardly present in any other registration. However, this study was able to include a large sample size of people in big and smaller cities across the country. Thanks to the participation of the national NSG network of street doctors and street nurses, we were able to gather information about the baseline characteristics and symptoms of COVID-19 -related issues among people experiencing homelessness.

It should be taken into account that the patients consulting street doctors are often passers-by and therefore street doctors regularly have no previous medical data on them. In addition, street doctor practices all have different registration systems for people experiencing homelessness and register different characteristics. Both were cause for a large amount of missing data. Due to the uncertain reach of the street doctors and the possibility that not all hospital admissions of people experiencing homelessness have been reported back to the street doctors, our data on hospital admissions might not be complete and possible show an underestimation.

There was a noticeable high number of missing data among the group "clinically suspected (but not tested)". A likely explanation is that most patients in this groups were registered in the first months of the pandemic, when there were no tests available and the whole situation was very hectic for street doctors. It is also possible that among this group, more people were not insured and tended to avoid care because of fear for costs.

Lastly, the response to the online questionnaire differed between the cities. This could be explained by the varying reach of the street doctors from NSG in the different cities. However, the response in the large cities of Amsterdam and Rotterdam was high, and it is plausible that the homeless population in these cities is comparable to the homeless population in other cities in the Netherlands. Despite these limitations, we believe we were able to capture a representative sample of this vulnerable population during the COVID-19 pandemic.

## Conclusion and recommendations

Although SARS-COV-2 infections were not as widespread among the homeless population in the Netherlands as expected, the number of hospitalizations among the group was relatively high, at a relatively young age, and may underscore the vulnerability of the homeless population and especially immigrants from outside the EU to a severe course of SARS-COV-2 infection. Monitoring this group during a pandemic is essential to be able to take the necessary actions to meet the care needs of people experiencing homelessness and improve accessibility of healthcare. The results of our study also indicate some weaknesses in the current healthcare system that could be improved to better prepare the homeless health care system for any future pandemic, such as the lack of a good registration or monitoring system. Monitoring and understanding the health, well-being and size of (at-risk groups within) the population of

people experiencing homelessness, as well as their use and need for healthcare, is structurally needed.

This is important not only to be better prepared for the next pandemic, but also to align current care and shelter with actual needs. In developing a monitoring system, one should acknowledge that the population is fluid and mobile and that there is much fear among the target population about capturing data about them [39].

## Acknowledgments

We thank the street doctors of the nine cities who participated in the study for making time to fill in the online registration forms.

## Author Contributions

**Conceptualization:** Maria van den Muijsenbergh, Tessa van Loenen.

**Data curation:** Tessa van Loenen.

**Formal analysis:** Eline Mennis, Michelle Hobus.

**Funding acquisition:** Maria van den Muijsenbergh.

**Investigation:** Eline Mennis, Michelle Hobus.

**Methodology:** Eline Mennis, Michelle Hobus, Tessa van Loenen.

**Project administration:** Tessa van Loenen.

**Supervision:** Maria van den Muijsenbergh, Tessa van Loenen.

**Validation:** Tessa van Loenen.

**Writing – original draft:** Eline Mennis, Michelle Hobus, Tessa van Loenen.

**Writing – review & editing:** Maria van den Muijsenbergh, Tessa van Loenen.

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
