## [Decision Letter · Decision Letter 0]

29 Mar 2023

PONE-D-23-02949COVID-19 related morbidity and mortality in homeless people in the NetherlandsPLOS ONE

Dear Dr. Van Loenen,

Thank you for submitting your manuscript to PLOS ONE. After careful consideration, we feel that it has merit but does not fully meet PLOS ONE’s publication criteria as it currently stands. Therefore, we invite you to submit a revised version of the manuscript that addresses the points raised during the review process.

We look forward to receiving your revised manuscript.

Kind regards,

Krit Pongpirul, MD, MPH, PhD.

Academic Editor

PLOS ONE

Journal Requirements: 

"Funding for this study was provided by the Netherlands Organisation for Health Research and Development (ZonMw) (https://www.zonmw.nl/en/), project number 10430022010005. ZonMw had no role in study design, data collection and analysis, decision to publish or preparation of the manuscript."

**Additional Editor Comments:**

Please carefully address the comments from both reviewers, especially the methodological concerns. Please also consider better word choices to describe the population as this is currently potentially stigmatising. 

Reviewers' comments:

Reviewer's Responses to Questions

**Comments to the Author**

1. Is the manuscript technically sound, and do the data support the conclusions?

Reviewer #1: No

Reviewer #2: Yes

2. Has the statistical analysis been performed appropriately and rigorously? 

Reviewer #1: Yes

Reviewer #2: Yes

3. Have the authors made all data underlying the findings in their manuscript fully available?

Reviewer #1: Yes

Reviewer #2: Yes

4. Is the manuscript presented in an intelligible fashion and written in standard English?

Reviewer #1: Yes

Reviewer #2: Yes

5. Review Comments to the Author

Reviewer #1: I assessed the manuscript by Mennis et al in which the authors aimed to assess morbidity and mortality of the COVID-19 virus among the homeless population in the Netherland. The authors collected data from street doctors during the COVID-19 period March 2020- March 2021 by means of an electronic repeated survey in which physicians were asked to report the medical encounters with homeless people seeking medical attention. The research question is of interest and in particular whether social minorities or not are at increased risk of SARS-CoV-2 infection and severe illness is still a matter of debate. Nevertheless, the study presents several limitations related to the study design which does not allow to catch clinical relevant outcome with enough precision (survey) and the study period comprehended the first wave in which SARS-CoV-2 testing was not available. In addition, no data regarding more recent period are lacking rendering the results poorly applicable in an actual situation of higher population exposure to previous infection and wide COVID-19 vaccine implementation.

Major comments:

• Methods: the main limitation rely in the study design definition. In particular, the authors define the study as a retrospective study and also as a survey. The mode of acquisition of the information is crucial to assess the potential bias and limitation of the study. In particular, the present study seems to be a repeated cross sectional survey provided to street doctors and thus not adequate in nature to assess outcomes such as disease severity, mortality and hospitalization (crude outcome) which requires a longitudinal study design (retrospective or prospective). Thus, the conclusion drawn by the authors seems not being supported by the study design. The only information that such a study could provide are related to which persons seek medical attention among homeless and the characteristics of such subjects at the time of medical encounter.

• Discussion line 224-225: this sentence is not supported by a study design able to assess if homeless are at increased risk of severe COVID-19 ouctome when compared to non homeless population.

Minor comments:

• SARS-CoV-2 instead of COVID-19 virus

• SARS-CoV-2 infection instead of COVID-19 infection

• Methods: definition of homeless is missing

• Methods: the definition of clinically suspected is missing

• Biological sex instead of sex

Reviewer #2: In this manuscript the Authors deal with the impact of COVID-19 pandemic on the homeless population in the Netherlands. It is a retrospective study in which anonymized data about homeless patients who contacted a street doctor were collected from March 2020 until March 2021 in 9 different Dutch cities.

The article is very well written and clear.

It gives a very interesting insight into this fragile population, and the street doctors did a remarkable job collecting the data during such a tough period. Besides, the method they used could be applied in further studies for collecting data about other diseases in the same or in other kind of populations of interest.

The study's strengths are that it is multicentric, it includes a wide number of subjects and the sample is representative and homogeneous.

Below you can find some suggestions to improve your article:

At the following lines, there are some extra brackets, which should be removed:

• line 51 (chronic)

• line 77 (outreaching)

• line 81 (COVID-19-related)

• line 287 (accessibility of)

• line 292 (risk groups within)

Line 111: “… in the five largest and 4 smaller cities in different parts of the Netherlands...”, it would be better to write both the number of cities either in numbers or in words.

Since the study started at the very beginning of the pandemic, it is recommended to clarify about covid-19 available literature to which the Authors refer at line 122.

Line 127: What is the meaning of the word “practices”? It is often found in the text, and it’s apparently referring to the doctors themselves. For example, what does the sentence cited above (line 127) mean? Does it mean that each doctor has his own registration system, or that each office/practice where they exercise has its system?

Lines 158-159: It is not clear how out of a total of 1544 consults, only 1419 homeless patients were registered, not even excluding the 114 patients who were seen more than once.

Line 196: “Most present in the group clinical suspected of COVID-19 were...” it would be better to reformulate the beginning of the sentence with something more appropriate.

Line 197 “Remarkable is that” needs to be changed with “it is remarkable that”.

From line 196 to line 200 the whole period needs to be rephrased.

Line 214 please, remove the commas between “One, migrant, patient”.

Line 221 please, rewrite the dates of the study period in numbers (March, 1st 2020/March, 1st 2021).

From line 290 to line 292 the whole period is confusing. Please, rephrase it.

6. PLOS authors have the option to publish the peer review history of their article (what does this mean?). If published, this will include your full peer review and any attached files.

Reviewer #1: No

Reviewer #2: **Yes: **Andrea Orsi

---

## [Author Response · Author response to Decision Letter 0]

15 May 2023

Dear Editor,

Thank you for your advice and offering us the opportunity to improve our manuscript with the valuable suggestions of the reviewers.

One of the comments is regarding the role of the funders. The following statement here applies: "The funders had no role in study design, data collection and analysis, decision to publish, or preparation of the manuscript." Thank you for offering to change this in our behalf. 

Another comment is regarding the data availability. There are ethical restrictions for sharing all of our data publicly. We researched people experiencing homelessness in a few Dutch cities. This is a relative small group in the Netherlands for which it is difficult for the results not to be indirectly traceable. Moreover, this is a vulnerable group which is why we want to be extra careful. For these reasons, we do not want to disclose the entire dataset. We have placed in the DANS repository all data that can be shared (e.g. summary findings, aggregates, and list of variable.) These can be found at in the DANS EASY, which is the preferred repository of our institution. Data can be found using the URL: Dakloosheid en Corona - EASY (knaw.nl) and DOI: 10.17026/dans-2c7-wksz.. In case of a request, we will have an ethical board look into which data can be shared. We hope this explanation will suffice. Thank you for offering to update the data availability statement. 

We hope we will have addressed the various comments to your satisfaction and you will deem our manuscript fit for publication.

Best regards,

On behalf of all co-authors,

Dr. Tessa van Loenen (corresponding author)

Journal Requirements: 

We have checked the templates again and hope we changed all necessary things accordingly. 

Thank you for noticing this, we will change this in the resubmission. 

"Funding for this study was provided by the Netherlands Organisation for Health Research and Development (ZonMw) (https://www.zonmw.nl/en/), project number 10430022010005. ZonMw had no role in study design, data collection and analysis, decision to publish, or preparation of the manuscript."

Indeed this statement is correct: "The funders had no role in study design, data collection and analysis, decision to publish, or preparation of the manuscript." 

As explained in our letter, ther are ethical restrictions for sharing all of our data publicly. We researched people experiencing homelessness in a few Dutch cities. This is a relative small group in the Netherlands for which it is difficult for the results not to be indirectly traceable. Moreover, this is a vulnerable group which is why we want to be extra careful. Given the delicate information and the easy to person retracable information, we do not want to disclose the entire dataset. We have placed in the DANS repository all data that can be shared (e.g. summary findings, aggregates, and list of variable.) These can be found at: DOI: 10.17026/dans-2c7-wksz. In addition, the data will be made available upon request. In case of a request, we will have an ethical board look into which data can be shared. We hope this explanation will suffice.

The repository in which we will place al data is: DOI: 10.17026/dans-2c7-wksz 

Additional Editor Comments:

#Please carefully address the comments from both reviewers, especially the methodological concerns. 

We addressed the methodological concerns in our reaction on the major comments of reviewer 1. 

#Please also consider better word choices to describe the population as this is currently potentially stigmatising. 

Thank you for warning us about possibly stigmatizing or offending word choice. As we ourselves are very much aware of the risk for and negative consequences of stigmatizing, we are very keen on avoiding this. However, probably our limit fluency in the English language has lead to unintended stigmatizing words.

The only one we could find is that we speak of “homeless” people, where the more correct phrasing would be “people experiencing homelessness “as their identity is not defined by their circumstances. We have corrected this throughout the manuscript.

Comments to the Author

Reviewer #1: I assessed the manuscript by Mennis et al in which the authors aimed to assess morbidity and mortality of the COVID-19 virus among the homeless population in the Netherland. The authors collected data from street doctors during the COVID-19 period March 2020- March 2021 by means of an electronic repeated survey in which physicians were asked to report the medical encounters with homeless people seeking medical attention. The research question is of interest and in particular whether social minorities or not are at increased risk of SARS-CoV-2 infection and severe illness is still a matter of debate. Nevertheless, the study presents several limitations related to the study design which does not allow to catch clinical relevant outcome with enough precision (survey) and the study period comprehended the first wave in which SARS-CoV-2 testing was not available. In addition, no data regarding more recent period are lacking rendering the results poorly applicable in an actual situation of higher population exposure to previous infection and wide COVID-19 vaccine implementation.

Dear reviewer,

We fully agree our study suffers from many limitations, which has to do with the limited means of assessing infections during the first waves of the pandemic as well as with the exceptional and challenging circumstances of providing healthcare for people who experience homelessness. We describe these limitations in our paper. Nevertheless, we think that, despite these limitations, our data provide some insight in an otherwise completely unknown territory, as the patients we describe are nowhere else registered or accounted for.

Major comments:

• Methods: the main limitation rely in the study design definition. In particular, the authors define the study as a retrospective study and also as a survey. The mode of acquisition of the information is crucial to assess the potential bias and limitation of the study. In particular, the present study seems to be a repeated cross sectional survey provided to street doctors and thus not adequate in nature to assess outcomes such as disease severity, mortality and hospitalization (crude outcome) which requires a longitudinal study design (retrospective or prospective). Thus, the conclusion drawn by the authors seems not being supported by the study design. The only information that such a study could provide are related to which persons seek medical attention among homeless and the characteristics of such subjects at the time of medical encounter.

Thank you very much for this comment. We clearly failed to write down our methodology with clarity causing confusion. Reading back, we understand where the confusion has occurred. in our opinion, it is mainly due to the word survey. So we believe that rewriting our method section will improve our manuscript. The reviewer describes our study as a cross sectional survey of healthcare providers but this is not the case. We would like to describe it as a retrospective analysis of streetdoctors anonymized registration data. We have tried to better describe the methodology in the hope that it is now clearer. We included the following in the manuscript to clarify:

Street doctors use different systems to record their medical data of patients and some do not have a data recording system for people who are homeless. Extracting the required data and comparing the data was not possible with the existing systems and processes. For this reason, we created a standardised registration form where street doctors could enter the relevant data from the consultation they had with each patient. This form was created using an online survey programme, Castor EDC. For each patient the street doctor saw, a form was filled in with to obtain required anonymous data. Once a month, the street doctors received a reminder to enter the data of the patients they had seen in those previous month. 

• Discussion line 224-225: this sentence is not supported by a study design able to assess if homeless are at increased risk of severe COVID-19 ouctome when compared to non homeless population.

You are right that we cannot prove this to be the case; so we have adjusted the sentence as followed:

Based on these numbers there seems to be no indication that COVID-19 among the group of people experiencing homelessness led to more morbidity than among peers, however we realize that our methods do not allow a conclusion on this comparison.

Minor comments:

• SARS-CoV-2 instead of COVID-19 virus

Changed accordingly 

• SARS-CoV-2 infection instead of COVID-19 infection

Changed accordingly 

• Methods: definition of homeless is missing

We included the following sentences in the method section on a definition of homelessness in the manuscript: People experiencing Homelessness are defined broadly as people who lack a steady home, and live in emergency shelters, outdoors, or in buildings not meant for shelter. This study includes all persons who use the service of a street doctor practice, whatever their sleeping place is.

• Methods: the definition of clinically suspected is missing

We added a definition in the method section: Patients were categorized as clinically suspected if they were diagnosed with COVID-19 but were not tested. Especially, at the beginning of the pandemic due to a shortage of PCR tests and policy guidelines, not all homeless people who contacted the street doctors could be tested. In addition, some homeless people refused to be tested. So both groups, were than considered clinically suspected of having COVID-19, by the street doctor, but were not tested.

• Biological sex instead of sex

Changed accordingly 

Reviewer #2: In this manuscript the Authors deal with the impact of COVID-19 pandemic on the homeless population in the Netherlands. It is a retrospective study in which anonymized data about homeless patients who contacted a street doctor were collected from March 2020 until March 2021 in 9 different Dutch cities.

The article is very well written and clear.

It gives a very interesting insight into this fragile population, and the street doctors did a remarkable job collecting the data during such a tough period. Besides, the method they used could be applied in further studies for collecting data about other diseases in the same or in other kind of populations of interest.

The study's strengths are that it is multicentric, it includes a wide number of subjects and the sample is representative and homogeneous.

Thank you so much for this positive comment. We really appreciate all the suggestions for changes and see that they make our manuscript much stronger.

Below you can find some suggestions to improve your article:

At the following lines, there are some extra brackets, which should be removed:

• line 51 (chronic)

• line 77 (outreaching)

• line 81 (COVID-19-related)

• line 287 (accessibility of)

• line 292 (risk groups within)

We removed the brackets accordingly 

#Line 111: “… in the five largest and 4 smaller cities in different parts of the Netherlands...”, it would be better to write both the number of cities either in numbers or in words.

Changed accordingly . 

#Since the study started at the very beginning of the pandemic, it is recommended to clarify about covid-19 available literature to which the Authors refer at line 122.

Thank you, indeed we started in a period that there was limited information available on specific symptoms. However, already in those early months it was clear that loss of smell, and gastro-enteric symptoms could indicate on covid-19 infection.

Since this part of the methodology was already changes because of comments of reviewer 1, we included a clarification of this in the revised version of the methodology. The senctence is refrased into: This form was created using an online survey programme Castor EDC and based on literature available in May 2020 about SARS-COV-2 and the expertise of experienced street doctors.

#Line 127: What is the meaning of the word “practices”? It is often found in the text, and it’s apparently referring to the doctors themselves. For example, what does the sentence cited above (line 127) mean? Does it mean that each doctor has his own registration system, or that each office/practice where they exercise has its system?

With the word practice we mean the general practices (offices) where the street doctors work – usually more doctors in 1 practice; each practice has its own registration system.

#Lines 158-159: It is not clear how out of a total of 1544 consults, only 1419 homeless patients were registered, not even excluding the 114 patients who were seen more than once.

Thank you, it is a very valid comment that it is unclearly described. We have explained the numbers more and added clarifying information. We rephrased into the following: 

There were a total of 1544 consultations for SARS-COV-2 related complaints or issues distributed among 1419 patients. 1305 of the patients came for one consultation and 114 patients came more than once (104 patients came 2 times, 9 patients came 3 times and 1 patient came 4 times)

#Line 196: “Most present in the group clinical suspected of COVID-19 were...” it would be better to reformulate the beginning of the sentence with something more appropriate.

We rephrased the sentence into: The most common symptoms patients in the group clinical suspected of SARS-COV-2 were a cold (66%), a cough (54%), a sore throat (35%), and shortness of breath (33%).

#Line 197 “Remarkable is that” needs to be changed with “it is remarkable that”.

Changed accordingly

#From line 196 to line 200 the whole period needs to be rephrased.

We rephrased into the following: 

The most common symptoms of patients in the group clinical suspected of COVID-19 were a cold (66%), a cough (54%), a sore throat (35%), and shortness of breath (33%). It was remarkable that in the group of patients that were clinically suspected of COVID-19 loss of taste and smell only occurred in 2%, whereas it was more common in the group that tested positive for COVID-19 (29%). Fever was reported more in the group ‘clinically suspected” (28%) than in the group was that tested negative for COVID-19 (13%).

#Line 214 please, remove the commas between “One, migrant, patient”.

We chanced the phrase into: one patient with a migrant background. 

# Line 221 please, rewrite the dates of the study period in numbers (March, 1st 2020/March, 1st 2021).

Changed accordingly 

From line 290 to line 292 the whole period is confusing. Please, rephrase it.

We agree that the sentence is confusion. We rephrased into the following: 

Monitoring and understanding the health, well-being and size of (at-risk groups within) the population of homeless people, as well as their use and need for healthcare, is structurally needed. This is important not only to be better prepared for the next pandemic, but also to align current care and shelter with actual needs.

---

## [Decision Letter · Decision Letter 1]

20 Oct 2023

PONE-D-23-02949R1SARS-COV-2  related morbidity and mortality in  people experiencing homelessness in the NetherlandsPLOS ONE

Dear Dr. Van Loenen,

Thank you for submitting your manuscript to PLOS ONE. After careful consideration, we feel that it has merit but does not fully meet PLOS ONE’s publication criteria as it currently stands. Therefore, we invite you to submit a revised version of the manuscript that addresses the points raised during the review process.

We look forward to receiving your revised manuscript.

Kind regards,

Krit Pongpirul, MD, MPH, PhD.

Academic Editor

PLOS ONE

Additional Editor Comments:

Please address the comments from both reviewers.

Reviewers' comments:

Reviewer's Responses to Questions

**Comments to the Author**

1. If the authors have adequately addressed your comments raised in a previous round of review and you feel that this manuscript is now acceptable for publication, you may indicate that here to bypass the “Comments to the Author” section, enter your conflict of interest statement in the “Confidential to Editor” section, and submit your "Accept" recommendation.

Reviewer #1: (No Response)

Reviewer #3: All comments have been addressed

2. Is the manuscript technically sound, and do the data support the conclusions?

Reviewer #1: Partly

Reviewer #3: Partly

3. Has the statistical analysis been performed appropriately and rigorously? 

Reviewer #1: Yes

Reviewer #3: Yes

4. Have the authors made all data underlying the findings in their manuscript fully available?

Reviewer #1: Yes

Reviewer #3: Yes

5. Is the manuscript presented in an intelligible fashion and written in standard English?

Reviewer #1: No

Reviewer #3: Yes

6. Review Comments to the Author

Reviewer #1: Abstract conclusions and conclusions of the manuscript "Although SARS-COV-2 infection was not widespread among people experiencing homelessness in the Netherlands, the number of hospitalizations in this study was relatively high compared to the general population."

this sentence is not supported by the study results becuase no "general population group" is provided as comparison.

Please use "SARS-CoV-2" when referring to the infection and "COVID-19" when referring to the disease.

Page 7 line 153-154 "2. Patients were categorized as clinically suspected if they were diagnosed with SARS-COV-2 but were not tested." please reword the sentence because it is not possible to diagnose SARS-CoV-2 infection without a molecular or antigenic test.

Reviewer #3: I have not reviewed the first version of the manuscript.

Comments regarding this revised version:

- Since the Castor EDC form contained information from literature up to May 2020, this form was apparently designed after May 2020. I understand very well that such form could not yet be made at the onset of the epidemic. However, for clarity it would be useful that the authors would mention at what moment the Castor form was introduced - and to state that all data regarding patients seen before that moment had to be added in retrospect. This makes it even clearer that data at the onset of the epidemic were probably scanty and it is understandable that many data are missing - although I do not see a clue why "biological sex" was missing so often.

A more fundamental comment that should have been raised regarding the first version of the manuscript is that there is no definition at all for "clinically suspected for SARS-CoV 2 infection". Therefore, the data as presented in table 3 regarding these groups will probably tell more about why the patients were put in this category. If clinicians, including street doctors, think that patients with fever are more likely to have SARS-CoV 2 infection, this will be reflected in these columns.

It is even less clear how patients were categorized as "clinically not suspected for SARS-CoV 2 infection". Apparently, this category includes both patients who presented with complaints after introduction of nationwide testing, but were not deemed elegible for testing by the clinician based on symptoms, and patients who presented with complaints before introduction of nationwide testing and might have been elegible for testing.

If I had reviewed the first version of the manuscript, I would have suggested to leave these categories -especially the last one- out of the manuscript. As it stands, it could also be sufficient to clarify these issues in the methods or in the result section.

7. PLOS authors have the option to publish the peer review history of their article (what does this mean?). If published, this will include your full peer review and any attached files.

Reviewer #1: No

Reviewer #3: No

---

## [Author Response · Author response to Decision Letter 1]

14 Nov 2023

Response to the reviewers

Reviewer #1: 

Dear reviewer, thank you for the valuable feedback. We have attempted to incorporate your suggestions into the manuscript as thoroughly as possible. Below, you will find point-by-point our response and modifications. 

• Abstract conclusions and conclusions of the manuscript "Although SARS-COV-2 infection was not widespread among people experiencing homelessness in the Netherlands, the number of hospitalizations in this study was relatively high compared to the general population."

this sentence is not supported by the study results because no "general population group" is provided as comparison.

We agree, we have changed both sentences and removed the part where we compare with general population. 

• Please use "SARS-CoV-2" when referring to the infection and "COVID-19" when referring to the disease.

Changed accordingly 

• Page 7 line 153-154 "2. Patients were categorized as clinically suspected if they were diagnosed with SARS-COV-2 but were not tested." please reword the sentence because it is not possible to diagnose SARS-CoV-2 infection without a molecular or antigenic test.

We have changed the sentence accordingly, emphasizing that diagnosis was not possible based on a molecular or antigenic test. We also added more information on how a diagnosis was made after a comment of reviewer #3. 

´Patients were categorized as clinically suspected if the street doctor diagnosed them with a SARS-CoV-2 infection, but confirmation through a molecular or antigenic test was not possible.”

Reviewer #3:

Dear reviewer, thank you for the valuable feedback. We have attempted to incorporate your suggestions into the manuscript as thoroughly as possible. Below, you will find point-by-point our response and modifications. 

• Since the Castor EDC form contained information from literature up to May 2020, this form was apparently designed after May 2020. I understand very well that such form could not yet be made at the onset of the epidemic. However, for clarity it would be useful that the authors would mention at what moment the Castor form was introduced - and to state that all data regarding patients seen before that moment had to be added in retrospect. This makes it even clearer that data at the onset of the epidemic were probably scanty and it is understandable that many data are missing - although I do not see a clue why "biological sex" was missing so often.

Thank you for this comment. We have added the information and agree that this is making it more clearer. 

“This form was created and finalized in May 2020 using an online survey programme Castor EDC. Items were based on literature available in May 2020 about COVID-19 and the expertise of experienced street doctors. For each patient the street doctor saw, a form was filled in with the required data. The data from March 2020 until May 2020 were added in retrospect. As of May 2020 , the street doctors received a reminder twice a month to enter the data of the patients they had seen in the previous two weeks.”

A more fundamental comment that should have been raised regarding the first version of the manuscript is that there is no definition at all for "clinically suspected for SARS-CoV 2 infection". Therefore, the data as presented in table 3 regarding these groups will probably tell more about why the patients were put in this category. If clinicians, including street doctors, think that patients with fever are more likely to have SARS-CoV 2 infection, this will be reflected in these columns.

It is even less clear how patients were categorized as "clinically not suspected for SARS-CoV 2 infection". Apparently, this category includes both patients who presented with complaints after introduction of nationwide testing, but were not deemed eligible for testing by the clinician based on symptoms, and patients who presented with complaints before introduction of nationwide testing and might have been eligible for testing.

If I had reviewed the first version of the manuscript, I would have suggested to leave these categories -especially the last one- out of the manuscript. As it stands, it could also be sufficient to clarify these issues in the methods or in the result section.

We understand this comment very well and have again tried to improve the definition of the various categories (especially clinically suspected and clinically not suspected). We have chosen to keep the categories in the text. Especially because omitting them would give an even more biased presentation. In the Netherlands in the first part of the pandemic there was no testing policy, and certainly for homeless population there was no testing capacity available or in some cases people refused a test. Street doctors therefore had to base their diagnosis on multiple facts and observations from history and taking physical examination. In those cases where all symptoms indicate to COVID And doctors often could or did not perform a test, classified the case as "clinically suspect." And on the other hand, when fever or other complaints were more likely explained by other conditions, they will have classified this as "not-suspected". we have made adjustments to the text so that hopefully the definitions are more clear. 

´Patients were categorized as clinically suspected if the street doctor diagnosed them with a SARS-CoV-2 infection, but confirmation through a molecular or antigenic test was not possible. Particularly at the beginning of the pandemic, due to a shortage of PCR tests and policy guidelines, not all individuals experiencing homelessness who sought assistance from street doctors could be tested. Additionally, some individuals declined testing. Consequently, street doctors had to rely on multiple facts and observations from medical history and physical examinations to make a diagnosis. Cases in which all symptoms pointed to COVID-19, and a PCR test could not or was not performed, were classified as "clinically suspect." Conversely, when fever or other complaints were more likely attributed to other conditions, these cases were categorized as "not suspected" of a SARS-CoV-2 infection”

---

## [Decision Letter · Decision Letter 2]

18 Dec 2023

COVID-19  related morbidity and mortality in  people experiencing homelessness in the Netherlands

PONE-D-23-02949R2

Dear Dr. Van Loenen,

We’re pleased to inform you that your manuscript has been judged scientifically suitable for publication and will be formally accepted for publication once it meets all outstanding technical requirements.

Kind regards,

Krit Pongpirul, MD, MPH, PhD.

Academic Editor

PLOS ONE

Additional Editor Comments (optional):

Your responses to the comments from both reviewers are satisfactory.

Reviewers' comments:

Reviewer's Responses to Questions

**Comments to the Author**

1. If the authors have adequately addressed your comments raised in a previous round of review and you feel that this manuscript is now acceptable for publication, you may indicate that here to bypass the “Comments to the Author” section, enter your conflict of interest statement in the “Confidential to Editor” section, and submit your "Accept" recommendation.

Reviewer #1: All comments have been addressed

2. Is the manuscript technically sound, and do the data support the conclusions?

Reviewer #1: Yes

3. Has the statistical analysis been performed appropriately and rigorously? 

Reviewer #1: Yes

4. Have the authors made all data underlying the findings in their manuscript fully available?

Reviewer #1: Yes

5. Is the manuscript presented in an intelligible fashion and written in standard English?

Reviewer #1: Yes

6. Review Comments to the Author

Reviewer #1: Although there are some methodological problems related to the case definition that could not be fixed the authors address an important topic on a specific often neglected population.

7. PLOS authors have the option to publish the peer review history of their article (what does this mean?). If published, this will include your full peer review and any attached files.

Reviewer #1: No

---

## [Editor Report · Acceptance letter]

26 Jan 2024

PONE-D-23-02949R2 

PLOS ONE

Dear Dr. Van Loenen, 

I'm pleased to inform you that your manuscript has been deemed suitable for publication in PLOS ONE. Congratulations! Your manuscript is now being handed over to our production team.

Kind regards, 

on behalf of

Assoc. Prof. Dr. Krit Pongpirul 

Academic Editor

PLOS ONE